# A Systematic Review of Prognostic Factors in Patients with Cancer Receiving Palliative Radiotherapy: Evidence-Based Recommendations

**DOI:** 10.3390/cancers16091654

**Published:** 2024-04-25

**Authors:** Alexander Tam, Emanuela Scarpi, Marco Cesare Maltoni, Romina Rossi, Alysa Fairchild, Kristopher Dennis, Marcus Vaska, Marc Kerba

**Affiliations:** 1Cumming School of Medicine, Department of Radiation Oncology, University of Calgary, Calgary, AB T2N 1N4, Canada; alexander.tam@ahs.ca; 2Unit of Biostatistics and Clinical Trials, IRCCS Istituto Romagnolo per lo Studio dei Tumori (IRST) “Dino Amadori”, 47014 Meldola, Italy; emanuela.scarpi@irst.emr.it; 3Medical Oncology Unit, Department of Medical and Surgical Sciences (DIMEC), University of Bologna, 40126 Bologna, Italy; marcocesare.maltoni@auslromagna.it; 4Palliative Care Unit, IRCCS Istituto Romagnolo per lo Studio dei Tumori (IRST) “Dino Amadori”, 47014 Meldola, Italy; romina.rossi@irst.emr.it; 5Department of Radiation Oncology, Cross Cancer Institute, Faculty of Medicine, University of Alberta, Edmonton, AB T6G 2R3, Canada; alysa.fairchild@ahs.ca; 6Division of Radiation Oncology, The Ottawa Hospital and the University of Ottawa, Ottawa, ON K1H 8L6, Canada; 7Knowledge Resource Service, Tom Baker Cancer Centre, Alberta Health Services, Calgary, AB T2N 4N2, Canada; marcus.vaska@ahs.ca

**Keywords:** cancer, advanced cancer, palliative, radiotherapy, prognostication, SBRT, re-irradiation, systematic review, prognostic factors, survival

## Abstract

**Simple Summary:**

Accurately predicting survival in patients with cancer receiving palliative radiotherapy is important for clinical decision making in cancer care management and delivery. This remains a challenge due to the heterogeneity of cancer diagnoses and a wide variety of prognostic factors. This study aims to review the literature to identify prognostic factors for clinical use as well as prognostic tools available to clinicians treating this population of patients. Based on the literature, we formulated evidence-based recommendations for clinicians to implement into practice with the intention of improving prognostic accuracy and overall patient care.

**Abstract:**

(1) Background: Prognostication in patients with cancer receiving palliative radiotherapy remains a challenge. To improve the process, we aim to identify prognostic factors in this population from the literature and offer evidence-based recommendations on prognostication in patients undergoing palliative radiotherapy for non-curable or advanced cancers. (2) Methods: A systematic review was performed on the medical literature from 2005 to 2023 to extract papers on the prognosis of palliative radiotherapy patients with advanced cancer. The initial selection was performed by at least two authors to determine study relevance to the target area. Studies were then classified based on type and evidence quality to determine final recommendations. (3) Results: The literature search returned 57 papers to be evaluated. Clinical and biological prognostic factors were identified from these papers to improve clinical decision making or construct prognostic models. Twenty prognostic models were identified for clinical use. There is moderate evidence supporting (i) evidence-based factors (patient, clinical, disease, and lab) in guiding decision making around palliative radiation; (ii) that certain biological factors are of importance; (iii) prognostication models in patients with advanced cancer; and that (iv) SBRT or re-irradiation use can be guided by predictions of survival by prognostic scores or clinicians. Patients with more favorable prognoses are generally better suited to SBRT or re-irradiation, and the use of prognostic models can aid in this decision making. (4) Conclusions: This evaluation has identified several factors or tools to aid in prognosis and clinical decision making. Future studies should aim to further validate these tools and factors in a clinical setting, including the leveraging of electronic medical records for data availability. To increase our understanding of how causal factors interact with palliative radiotherapy, future studies should also examine and include prediction of response to radiation as an outcome.

## 1. Introduction

Accurately determining cancer prognosis can influence the delivery of quality care for patients with advanced cancer. The prediction of a patient’s likely survival informs clinical decision-making as well as the selection of appropriate treatments, and allows patients and their caregivers to plan accordingly [1]. Inaccurate prognoses can lead to futile treatment and undue suffering for the patient, decreasing their quality of life and adversely impacting their care. This is especially important as patients approach end of life (EoL) and enter palliative care [2]. It is also relevant to determining who may benefit from palliative radiotherapy (PRT). PRT is the use of radiation to alleviate pain and other symptoms or control disease without curative intent. Prognoses must be accurate during this phase of care to maximize the duration and quality of life while minimizing the time spent actively receiving treatment and the toxicity from treatment.

Developing an accurate prognosis also allows patients to make better-informed decisions about their remaining course of life, such as when to focus on symptom control or avoid hospitalization/acute care interventions. Historically, the use of prognostication in clinical practice has been left to individual clinician predictions based on experience and patient trends. Unfortunately, this method is heavily subjective and inaccurate, leading to inaccurate estimates of patient survival [1]. These inaccuracies can also lead to decreased clinician confidence in communicating information about a diagnosis.

With the advent of big data and the analysis of EMR data, prognostic factors are now more readily identified, and can be better studied and validated for use in guiding prognoses. Either individually or considered as part of a broader model, these prognostic factors can greatly improve the accuracy and reliability of prognostication in advanced cancer patients referred for PRT [3].

### Key Questions

This study aims to review the literature on prognostic factors in PRT as well as the prognostic tools for patients receiving PRT, and determine how this literature should be translated to inform recommendations for clinical practice. In patients with advanced cancer, how does the current evidence guide the use of PRT, the integration of estimates of life-expectancy, and the use of prognostic tools/predictive tools and their implementation in clinical practice?

## 2. Materials and Methods

### 2.1. Identifying Target Population

The target population of this review are patients with advanced cancer referred for palliative radiation. Palliative radiation is defined as non-curative radiation meant for symptom or pain relief. Palliative radiation does not preclude SBRT, retreatment, or complex radiotherapy plans where appropriate.

### 2.2. Systematic Literature Search

A systematic review was performed for the area of interest on the MEDLINE (Ovid), EMBASE, and PubMed databases. Given the heterogeneity of the data and the paucity of RCTs in this area, a meta-analysis of the relevant literature is beyond the scope of this review.

A search of the scientific literature was undertaken for the period from Jan 2005 to Dec 2023. Search terms included the following: decision-making, adult patients with cancer, radiotherapy, prognosis. Exclusion criteria included the following: reviews, commentaries, letters, news articles, non-English, non-palliative radiotherapy, curative intent radiotherapy. A detailed table outlining the search methodology is included in the Appendix A. This review was performed in accordance with the PRISMA (Preferred Reporting Items for Systematic Reviews and Meta-Analyses) guidelines and has not been registered.

### 2.3. Assigning the Level of Evidence to the Selected Literature

Each original study underwent preliminary evaluation by at least four group members to determine relevance to the area of interest. Studies were then classified by study quality (Figure 1 [3]) and type using the classification method (Figure 2 [4]) described by Maltoni et al. and the Oxford Evidence-Based Medicine Levels of Evidence grading system.

### 2.4. Formulating and Grading Final Recommendations

Final decisions on study eligibility were made based on specific relevance to the defined key questions. Recommendations were formulated from the selected studies based on key questions and patterns within the literature. Any disagreements were resolved by consensus among the four co-evaluators/co-authors.

GRADE quality assessment recommendations were then assigned to each outcome in collaboration with the co-authors as per Figure 3, with final recommendations graded using the modified GRADE scoring system [5]. We used a modified GRADE scoring scale to rate the strength of the recommendations (strong, weak, moderate). Where strong and weak from the grade tool did not apply, we used moderate to reflect uncertainty. Where recommendations were not dichotomized into strong or weak, we used moderate as an alternative to conditional, recognizing the validity of that rating in the GRADE system [5]. This requirement may be a function of this research area and the inherent uncertainty around prognostication/methods used to determine the validity of models/predictions.

## 3. Results

Our search returned 430 citations in total (after duplicates were removed). A PRISMA diagram is included in Figure 4. Studies selected as evidence are listed in Table 1.

### 3.1. Recommendations

#### 3.1.1. Recommendation 1: Quality: Moderate; Strength: Moderate

The clinical decision to recommend PRT may benefit from the use of evidence-based prognostic factors to guide decision-making.

Patient factors, disease factors, and lab factors were found in 30 studies to contribute to informed decision making in the palliative setting for patients receiving PRT. Large studies by Ma and Zaorsky established nomograms that could be used to guide decisions by reliably identifying subgroups of patients with poor prognosis [29,33]. High-quality evidence in the bone metastases population was equally able to differentiate among groups of patients receiving pRT based on clinical factors into categories that differentiated survival and could allow for avoidance of under- and overtreatment [7,25]. Westhoff et al. built upon the work from the Dutch Bone Metastasis study to develop a clinically useful model to predict survival in patients with painful bone metastases using sex, primary tumor, visceral metastases, KPS, and scales of general health and life with a c stat of 0.72 [16]. Studies by Nieder et al. modelled survival, noting the importance of ECOG in predicting survival during pRT, and also identified a different value of ESAS-based scoring as a method of identifying patients with different survivals [20,24].

Overall, we identified 35 clinically obtainable factors to guide prognosis in the PRT setting. The patient factors of age, KPS/ECOG, ESAS symptoms (pain, tiredness, drowsiness, nausea, shortness of breath, appetite, depression, anxiety, and wellbeing), cachexia, sarcopenia, prior hospitalizations, Charlson comorbidity index, opioid/steroid needs, and pleural effusion were identified to be significant in one or more studies [6,7,8,9,11,12,13,14,15,16,17,18,19,20,21,22,23,24,25,26,27,28,29,30,31,32,33,34,35,37]. These factors are readily available to prescribing clinicians and can give insight into a patient’s survival progression. Disease and treatment factors were also validated as prognostically significant in patients with advanced cancer prescribed PRT. Patient factors (17) identified in the review included the following: age, KPS/ECOG, ESAS symptoms (pain, tiredness, drowsiness, nausea, shortness of breath, appetite, depression, anxiety, and well-being), cachexia, sarcopenia, prior hospitalizations, Charlson comorbidity index, opioid analgesics/steroid needs, and pleural effusion [6,7,8,9,11,12,13,14,15,16,17,18,19,20,21,22,23,24,25,26,27,28,29,30,31,32,33,34,35,37]. Recognized disease and treatment factors (14) included the following: type of cancer, # of diagnoses, systemic therapy, disease-free interval, radiation dose, disease progression, primary tumor characteristics grade, histology, T stage, N stage, site and size of metastases (bone, lung, hepatic, adrenal gland, brain), site of surgery, tumor thrombus (portal vein) [6,7,8,9,11,12,13,14,15,16,17,18,19,20,21,22,23,24,25,26,27,28,29,30,31,32,33,34,35,37]. Additionally, common lab factors (4) were identified as prognostically significant in one or more studies, including lactate dehydrogenase, WBC (leukocytosis), hemoglobin (anemia), and C-reactive protein levels [18,20,21,23,26].

The identification and validation of how these factors may be used as prognostic tools is important as they allow for an improved patient selection of those who would likely benefit from the prescription of PRT. By better understanding the disease and treatment, clinicians can also more readily communicate diagnoses and use these factors either independently or as part of a model to aid in the prediction of survival.

#### 3.1.2. Recommendation 2: Quality: Moderate; Strength: Moderate

Certain biological factors appear to be significant in the prognosis of certain diseases.

There was a limited amount of work that identified biological factors that could contribute to informed decision making in the palliative setting for patients receiving PRT. The methylated MGMT promotor was noted by Zwirner et al. in a study population of n = 51 patients receiving re-irradiation for glioblastoma with disease progression that approached statistical significance. Well-known prognostic factors including the use of TMZ with the initial RT only appeared relevant in a subgroup of four long-term survivors [22]. Work by Hua et al. in a retrospective study of PRT in advanced liver cancer showed that alpha-fetoprotein (HR = 2.098) modelling had a potential to predict survival [32]. This was independent of other clinical factors and the dose of RT itself. There is limited evidence to guide the use of these factors outside the noted diseases. They may still, however, be considered as prognostic tools to identify patients who would benefit from the prescription of PRT treatment.

#### 3.1.3. Recommendation 3: Quality: Moderate; Strength: Moderate

Prognostic models and tools developed for use in patients with advanced cancer can be used in the decision to prescribe PRT.

Nineteen studies were identified that created nomograms or had decision-making scoring tools for use by clinicians to better prognosticate patients (Table 2). The multivariate analysis of collected prognostic factors determined significance and corresponding hazard ratios to construct the final models. Additionally, decision-making trees and recursive partitioning techniques were also used to develop novel clinical tools to better determine patient survival.

Work by Westhoff et al. (grade 2), used the findings from the Dutch Bone Metastasis study to develop a clinically useful model to predict survival in patients with painful bone metastases using sex, primary tumor, visceral metastases, KPS, and scales of general health and life with a reasonable c stat of 0.72 [16]. Other works carried out by Combs, Bollen, Katagiri, and Willeumier developed tools to be used in patients with bone and spinal metastases [6,12,15,25].

Gensheimer, in 2019, leveraged the data contained in the EMRs of more than 12,500 patients, including PRT courses and the prediction of overall survival (OS) [28]. For PRT, the C-index was 0.745 for OS. Zaorksy, in 2021, established the METSSS model of over 68,500 patients as a prognostic tool for the overall survival of cancer patients after PRT [33]. This work was further validated by Christ et al., focusing on metastatic patients post-PRT [41]. This involved calculating a mortality risk score followed by the stratification of all patients to prognostic risk groups, and correctly predicted the survival of end-of-life patients.

The studies detailing the decision-making tools noted in Table 2 can be used to better determine patients that may benefit from PRT. Further studies should help validate these models across palliative settings, including patient and disease factors. In the interim, clinicians can familiarize themselves further with these models and how they improve prognostic accuracy. Highly specialized models may prove especially useful for certain advanced cancers, as they appear to include specialized, impactful prognostic factors that increase accuracy. While these tools should not be the sole method of prognosis, they offer guidance to oncologists and other health providers to integrate with their own clinical experience.

#### 3.1.4. Recommendation 4: Quality: Moderate; Strength: Moderate

SBRT/Re-irradiation decision making.

Our study identified 12 papers that included a discussion of stereotactic body radiation therapy (SBRT) and re-irradiation in patients with non-curable cancer. Franzese et al. reviewed SBRT in patients across three Italian centers with adrenal gland metastases [31]. Their findings demonstrated that SBRT toxicity was low and the method was effective in proving local control. In the setting of brain metastases, Leth et al. examined a retrospective cohort of n = 198 patients and identified four prognostic factors related to OS after SRS [19]. In a multivariate analysis, they identified age ≥ 65 years, performance status ≥ 2, extracranial metastases and size of metastasis >20 mm as independent factors related to shorter survival and recommended that patients with three or four factors not be offered SRS. Kessel et al. examined the use of PRT and validated the Combs prognostic index among n = 199 patients with recurrent glioma [12,39]. It was suggested based on their limited analysis and noted survival among the risk groups that the Combs Prognostic Score should be used in clinical decision making and patient stratification.

We observed that in select situations, such as recurrent disease, brain metastases, and spinal cord compression, there is some evidence to guide who may be suitable for retreatment or SBRT based on their life expectancy ([11,19,22,28,31,32,34,35,38,39,43,44] and [12] (p. 20)). Patients with a longer predicted survival are generally more suitable for longer courses of PRT, as they are more likely to complete their treatment [45]. The use of SBRT, as demonstrated in SC24 [46], is thus likely more appropriate in patients with longer predicted survivals as well. Prognostic tools can be used to identify these patients and aid clinicians in decision making.

## 4. Discussion

Our study was concerned with offering evidence-based recommendations on prognostication in patients undergoing PRT for non-curable or advanced cancers and our findings and recommendations are limited to this PRT population. Other studies have examined how to utilize prognostication in more generalized populations with advanced cancers [2]. Studies have shown that up to 10% of patients prescribed PRT do not complete their treatment course [44,45]. Better prognostication and a more judicious selection of who may benefit from PRT should improve QoL for patients and resource utilization in departments with constrained resources.

We observed that retrospective validation was used in the majority of PRT studies. While preliminary validation studies are complete, a remaining challenge is how to integrate models and prognostic tools into clinical practice and how to evaluate outcomes resulting from changes in practice. A lack of information about the performance of models in practice is a factor limiting the applicability of these studies and preventing the widespread translation of research findings to clinical care. This limits our selection and recommendation of a specific model to be used in prognostication.

In the SRS and SBRT setting, we are mindful of studies examining the role of radiation to treat oligometastases and brain metastases that straddle the palliative to radical intent spectrum. Many contemporary studies were thus not included in this systematic review as they were outside the scope of our current work. We also observed a limitation in the quantity of non-common, laboratory, biological factors described in studies associated with prognostication in PRT. Future work should look to further explore how these factors may contribute to prognostic tools in pRT across various cancers.

The era of big data, AI, and the proliferation of EMRs can facilitate the development and integration of prognostication models and algorithms into clinical workflows. As these tools mature, accessing them seamlessly through EMRs could improve the quality of cancer care and workflows within the RT department. While prognostication is far from being an automated process, clinician judgement coupled with prognostic modelling could increase the accuracy and reliability of prognoses within advanced cancer care.

In summary, most studies focus on predicting overall survival to aid in clinical decision making. There is limited research examining the prediction of responses to the prescribed therapies from these data. We suggest that future studies also better identify prognostic factors that predict response to these therapies in addition to overall survival. Response could be represented by disease control or symptom status in patients prescribed palliative radiotherapy. This should better determine those patients that may respond to an intensification of treatment better than others [44].

## 5. Conclusions

Prognostication in patients with advanced cancer undergoing palliative radiotherapy remains a challenge. This evaluation has identified that several factors or tools can be used to aid in prognosis and clinical decision making. Future studies should aim to further validate these tools and factors in a clinical setting, including the leveraging of electronic medical records for data availability. To increase our understanding of how causal factors interact with palliative radiotherapy, future studies should also examine and include prediction of response to radiation as an outcome.

## Figures and Tables

**Figure 1 cancers-16-01654-f001:**
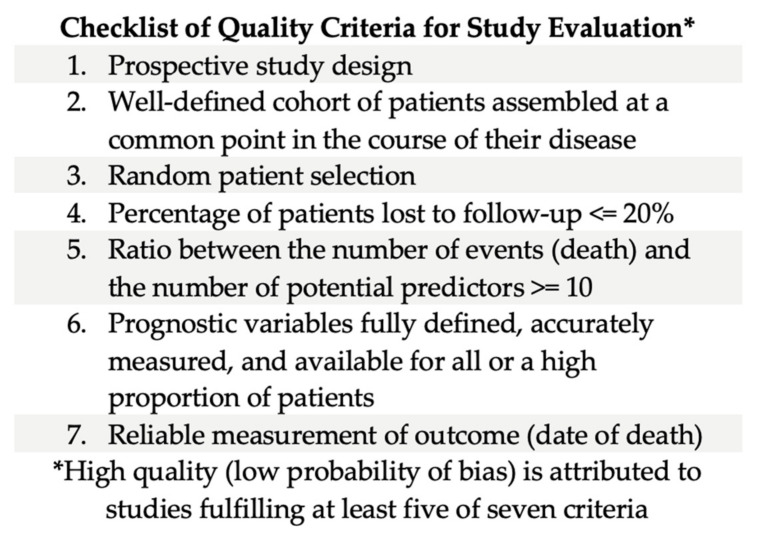
Quality criteria checklist [3].

**Figure 2 cancers-16-01654-f002:**
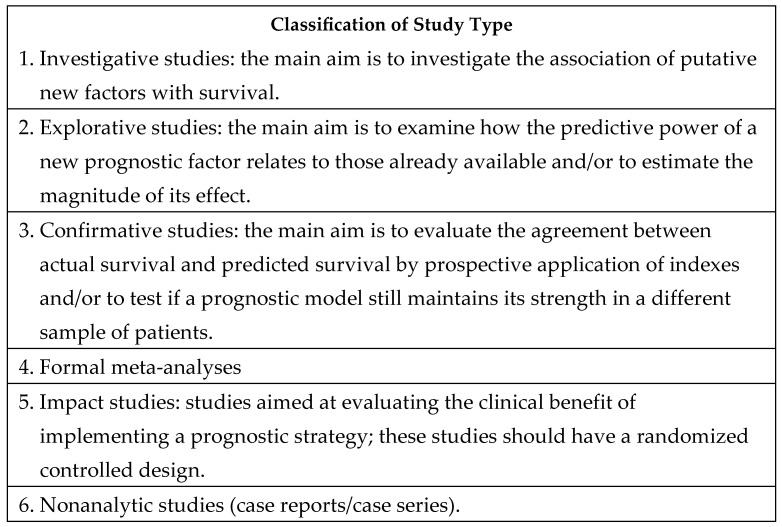
Classification of study type [3].

**Figure 3 cancers-16-01654-f003:**
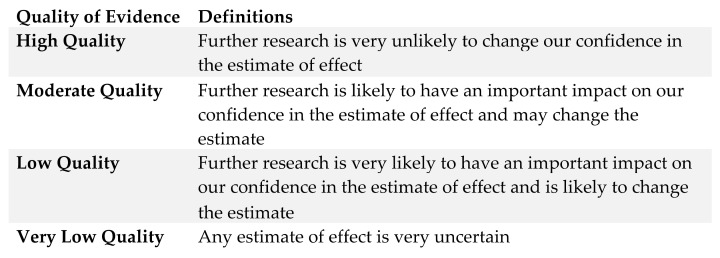
Modified GRADE scoring system [5].

**Figure 4 cancers-16-01654-f004:**
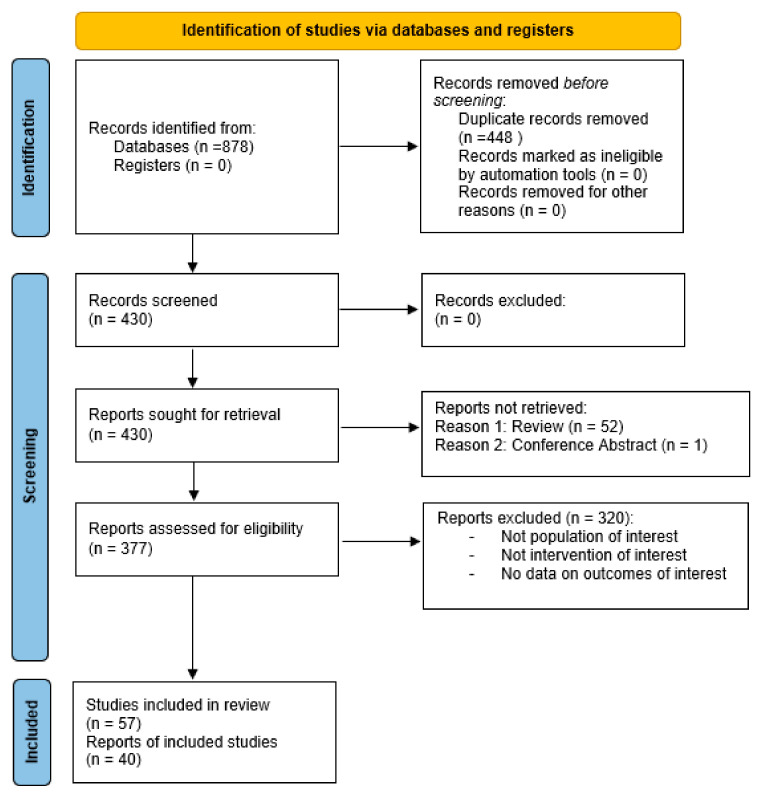
PRISMA diagram for prognostication in palliative radiotherapy.

**Table 1 cancers-16-01654-t001:** Results of literature review: summary of studies used to formulate recommendations.

Area Considered	No. of Articles Selected	Selected Studies		No. of Pts	Criteria Quality Checklist Filled [3]	Study Classification Type [3]	Evidence Grade [4]
		Reference	Year				
Guiding clinical decision making (prediction of survival for RT suitability)	30	Katagiri [6]	2005	350	6	2	4
		Van der Linden [7]	2005	342	5	2	2
		Chow [8]	2008	1307	5	2–3	3
		Mizumoto [9]	2008	544	5	2	3
		Chow [10]	2009	1307	5	2–3	3
		Rades [11]	2011	382	3	2	3
		Combs [12]	2012	233	5	2–3	3
		Krishnan [13]	2014	862	5	2–3	3
		Nieder [14]	2014	539	4	2	3
		Bollen [15]	2014	1385	5	2–3	3
		Westhoff [16]	2014	2091	5	2–3	2
		Oorschot [17]	2014	120	4	2	4
		Angelo [18]	2014	412	5	3	3
		Leth [19]	2015	198	5	2	4
		Nieder [20]	2015	873	5	2	3
		Nieder [21]	2016	781	5	2	3
		Zwirner [22]	2016	51	5	2	5
		Nieder [23]	2018	232	4	2	3
		Nieder [24]	2018	94	4	1–2	4
		Willeumier [25]	2018	1750	5	2–3	3
		Lorenzo [26]	2018	99	4	2	3
		Syadwa [27]	2018	585	5	2	3
		Gensheimer [28]	2019	12,987	4	2–3	3
		Ma [29]	2019	1593	5	2–3	3
		Yao [30]	2019	234	5	2–3	3
		Franzese [31]	2021	142	5	2	4
		Hua [32]	2021	159	4	2–3	3
		Zaorsky [33]	2021	68,505	5	2–3	3
		Mori [34]	2022	304	5	2	3
		Walker [35]	2022	269	5	2	3
Biological factors that influence prognosis	2	Zwirner [22]	2016	51	5	2	5
		Hua [32]	2021	159	4	2–3	3
Prognostic tools	19	Katagiri [6]	2005	350	6	2	4
		Van der Linden [7]	2005	342	5	2	2
		Chow [8]	2008	1307	5	2–3	3
		Mizumoto [9]	2008	544	5	2	3
		Chow [10]	2009	1307	5	2–3	3
		Combs [12]	2012	233	5	2–3	3
		Angelo [36]	2014	412	5	3	3
		Krishnan [13]	2014	862	5	2–3	3
		Bollen [15]	2014	1043	5	2–3	3
		Westhoff [16]	2014	1157	5	2–3	2
		Nieder [23]	2018	232	4	2	3
		Willeumier [25]	2018	1520	5	2–3	3
		Lorenzo [26]	2018	99	4	2	3
		Gensheimer [28]	2019	12,987	4	2–3	3
		Ma [29]	2019	1593	5	2–3	3
		Yao [30]	2019	234	5	2–3	3
		Hua [32]	2021	159	4	2–3	3
		Zaorsky [33]	2021	68,505	5	2–3	3
		Walker [35]	2022	269	5	2	3
Validation	8	Chow [37]	2009	445	5	3	3
		Angelo [36]	2014	412	5	3	3
		Buergy [38]	2016	52	4	3	4
		Kessel [39]	2017	199	5	3	3
		Kain [40]	2020	862	5	3	3
		Christ [41]	2022	274	5	3	3
		Maltoni [42]	2022	255	6	3	3
		Sakurai [43]	2022	485	5	3	3
SBRT/Re-irradiation	12	Rades [11]	2011	191	5	2	3
		Combs [12]	2012	233	5	2–3	3
		Steinmann [44]	2012	151	6	2	3
		Leth [19]	2015	198	5	2	4
		Zwirner [22]	2016	51	5	2	5
		Buergy [38]	2016	52	4	3	4
		Kessel [39]	2017	199	5	3	3
		Gensheimer [28]	2019	12,987	4	2–3	3
		Franzese [31]	2021	142	5	2	4
		Hua [32]	2021	159	4	2–3	3
		Walker [35]	2022	269	5	2	3
		Sakurai [43]	2022	485	5	3	3

**Table 2 cancers-16-01654-t002:** Summary of studies of prognostic models and tools for patients receiving palliative RT.

Reference	Disease Site	Pts (n =)	RT Details (RT Treatment Type */ Retreatment)	Prediction Forecast	Prognostic Factors	Model Results and Accuracy	Validating Studies
Treatment Site: Bone							
Bollen [15]	Symptomatic spinal metastases	1043	Unspecified Unspecified	<36 Months	Primary tumor, clinical profile, performance status, presence of visceral/brain mets	Four groups based on predictive model using prognostic factor weights. Median OS in months (31.2, 15.4, 4.8, 1.6). C Stat: 0.69.	
Katagiri [6]	Skeletal metastases	350	Unspecified Unspecified	<12 Months	Site of primary lesion, performance status, presence of visceral/brain metastases, previous chemotherapy, multiple skeletal metastases	Predictive model using prognostic factor weights. % likelihood of survival after 6 months based on scoring system (98%, 31%).	Sakurai 2022 [43]
Mizumoto [9]	Spinal metastases	544	Unspecified Unspecified	<24 Months	Unfavourable tumor type, bad performance status, hypercalcemia, visceral metastases, previous chemotherapy, multiple bone metastases, age >71	Three groups based on predictive model using prognostic factor weights. Median OS in months (27.1, 5.4, 1.8).	
Van der Linden [7]	Symptomatic spinal metastases	342	Unspecified No	<24 months	KPS, primary tumor, visceral metastases	Three groups based on predictive model using prognostic factor weights. Median OS in months (3.0, 9.0, 18.7).	
Walker [34]	Spinal metastases	269	CRT Including Re-irradiation	<12 months	KPS, histology, stability of disease	Three groups based on predictive model using prognostic factor weights. Median OS in months (11.4, 6.3, 2.0).	
Westhoff [16]	Symptomatic bone metastases	1157	Unspecified Unspecified	<24 months	Sex, primary tumor, visceral mets, KPS, visual analog scale general health, valuation of life verbal rating scale	Predictive model using prognostic factor weights. Median OS 21 weeks. C stat: 0.72.	
Willeumier [25]	Symptomatic long bone mets	1770	Unspecified Unspecified	<24 months	Clinical profile, KPS, evidence of visceral/brain met, solitary bone metastasis, and sex	Four groups based on predictive model using prognostic factor weights. Median OS in months (21.9, 10.5, 4.6, 2.2). C stat. 0.70.	
Brain							
Yao [30]	Bladder cancer with brain metastases	468	Unspecified Unspecified	<9 months	Brain metastasis, surgery of the primary site, chemotherapy, radiation therapy, palliative care, brain confinement of metastatic sites, and the Charlson/Deyo score	Predictive model using prognostic factor weights. High- and low-risk groups based on model. Median OS in months (1.68, 8.05), respectively. AUC for 0.5- and 1-year survival (0.838, 0.809), respectively	
Multiple Sites							
Angelo [18]	Metastatic/ incurable cancer	412	Unspecified Unspecified	<1 month	ECOG PS 3–4, opioid analgesic use, low Hb, steroid use, known progressive disease outside PRT target volume	RPA classification tool using prognostic factor weights. Median OS 6.3 months. Model correctly identified 75% of PRT courses administered during the final 30 days of life.	
Chow [10]	Advanced cancer	1308	Unspecified Unspecified	<12 months	KPS, interval from diagnosis, analgesic consumption, ESAS symptoms	Three groups based on RPA classification tool using prognostic factor weights. Median OS in weeks (32, 23, 11).	
Chow [8]	Metastatic cancer	1307	Unspecified Unspecified	<18 months	Non-breast cancer, metastases other than bone, KPS < 60	Three groups based on predictive model using prognostic factor weights. Median OS in weeks (64, 29, 10). C stat: 0.63	Chow 2009 [10]
Chow [8]	Metastatic cancer	1307	Unspecified Unspecified	<18 months	Non-breast cancer, metastases other than bone, KPS < 60	Three groups based on predictive model using number of prognostic factors. Median OS in weeks (64, 29, 10). C stat: 0.63.	Sakurai 2022 [43]
Gensheimer [28]	Metastatic cancer	12,987	CRT/SBRT/SRS Unspecified	< 12 months	Fully automatic (4126 variables)	Predictive model using automated prognostic factor weights. Median OS 20.9 months C stat: 0.745	
Krishnan [13]	Metastatic cancer	862	Unspecified No	< 24 months	Type of cancer, ECOG, age, prior palliative chemotherapy, prior hospitalizations, and hepatic metastases	Three groups based on predictive model using number of prognostic factors. Median OS in months (19.9, 5, 1.7).	Kain 2020 [40], Maltoni 2022 [42]
Zaorsky [33]	Metastatic cancer	68,505	Unspecified Unspecified	< 48 months	Metastasis location, age, primary tumor, gender, Charlson comorbidity score, RT site	Three groups based on predictive model using prognostic factor weights. Median OS in months (11.66, 5.09, 3.28). C stat: 0.71.	Christ 2022 [41]
Other Sites							
Combs [12]	Recurrent glioma	233	SRS Inclusive of re-irradiation	< 24 months	Histology, age, time between initial RT and re-irradiation	Four groups based on predictive model using prognostic factor weights. % likelihood of survival after 6 months (89%, 82%, 68%, 70%).	Kessel 2017 [39]
Hua [32]	Advanced liver cancer	159	CRT/SBRT Unspecified	< 24 months	Bone metastasis, portal vein tumor thrombus, alpha-fetoprotein, radiation dose	Predictive model using prognostic factor weights. Median OS 14.8 months. C stat: 0.735.	
Lorenzo [26]	Metastatic uveal melanoma	99	Unspecified Unspecified	<12 months	Age > 65, lactate dehydrogenase, size of liver metastasis, gamma glutamyl transpeptidase	RPA classification tool based on prognostic factor weights. Two survival patterns observed (>12 months, <12 months).	
Ma [29]	Metastatic gastric adenocarcinoma	1593	Unspecified Unspecified	<40 months	Age, tumor size, location, grade, T stage, N stage, metastatic site, scope of gastrectomy, number of examined lymph node(s), chemotherapy and radiotherapy	Two groups based on predictive model using prognostic factor weights. Median OS in months (5.0, 12.0). C stat (Pre-Operative): 0.607. C stat (Post-Operative): 0.699.	
Nieder [23]	Lung cancer	232	Unspecified No	<12 months	Performance status, lactate dehydrogenase, C-reactive protein, liver/adrenal gland metastases, and extrathoracic disease status	Four groups based on predictive model using prognostic factor weights. Median OS in months: (0.8, 1.6, 3.3, 10.5).	

* RT type: CRT, SBRT, SRS, or unspecified.

## Data Availability

The original contributions presented in the study are included in the article/Appendix A; further inquiries can be directed to the corresponding author/s.

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
