# Peer review of "A Systematic Review of Prognostic Factors in Patients with Cancer Receiving Palliative Radiotherapy: Evidence-Based Recommendations"

_cancers, 2024, doi:10.3390/cancers16091654_

Round 1
Reviewer 1 Report
Comments and Suggestions for Authors
I read this article about the role of palliative irradiation in a prognostic optical with great interest.
Analysing a topic that lacks clear guidelines is certainly interesting and the effort to conduct a systematic review of the literature is commendable.
Table 2 is difficult to read. Can the authors divide it into two (or more?) and highlight the type of RT treatment performed?
To improve clarity, it may be useful to categorise treatments by site. Brain metastases, for example, require a different procedure.
Additionally, it would be valuable to have partial results regarding the prognostic definition.
Comments on the Quality of English Language
English is adequate.
Author Response
Thank-you for your review.
We have made some minor revisions to improve the clarity of the manuscript findings as requested by;
- Updating Table 2 to better communicate our findings. As requested, we have expanded on the factors involved in the delivery of RT, the prognostication modelling, describing some of their results and organizing the results overall by treatment site.
Sincerely,
The authors
Reviewer 2 Report
Comments and Suggestions for Authors
1. The manuscript consists of a systemic review of palliative radiotherapy with reference to prognostic factors to inform clinical decision making.
2. Labeling tables and figures at the top is preferable to placing the label at the bottom.
3. Consider placing tables 1 and 2 in the appendex.
4. Section 2.4 concerning the formulating and grading of final recommendations–how did the authors come to an agreement?
5. Include a table of significant findings contributing to inform decision making and prognostication and the results such as:
1. Patient factors
2. Disease factors
3. Treatment factors
4. Availability of prognostic tools and indices
5. Outcomes measures
Author Response
Thank-you for your review.
We have made some minor revisions to improve the clarity of the manuscript by addressing the concerns described by;
- Now labeling tables and figures at the top (this was an oversight on our part).
- Given the consensus among reviewers and our preference for communicating the findings, we have maintained Tables 1 and 2 in the body of the manuscript. Thank-you for your suggestion.
- We have added a statement in Section 2.4 stating that an agreement was derived by consensus of the 4 co-evaluators/co-authors.
- We have updated Table 2 to better communicate our findings (rather than create a new table) by expanding on the factors involved in the prognostication modelling, describing their results/outcomes and organizing the results by treatment site.
Sincerely,
The authors
Reviewer 3 Report
Comments and Suggestions for Authors
Dear author,
The title of your systematic review, "A Systematic Review of Prognostic Factors in Patients with Cancer Receiving Palliative Radiotherapy: Evidence-based Recommendations," clearly outlines the scope and purpose of your study. This is crucial for readers to understand what to expect from your review.
In my opinion, your systematic review will serve as a valuable resource for clinicians and researchers seeking evidence-based guidance in palliative radiotherapy for cancer patients.
Author Response
Thank-you for your review.
We have made some minor revisions to improve the clarity of the manuscript by addressing some of the concerns around Table 2 as requested by your co-reviewers.
Sincerely,
The authors